# Feature Extraction of a Non-Stationary Seismic–Acoustic Signal Using a High-Resolution Dyadic Spectrogram

**DOI:** 10.3390/s23136051

**Published:** 2023-06-30

**Authors:** Diego Seuret-Jiménez, Eduardo Trutié-Carrero, José Manuel Nieto-Jalil, Erick Daniel García-Aquino, Lorena Díaz-González, Laura Carballo-Sigler, Daily Quintana-Fuentes, Luis Manuel Gaggero-Sager

**Affiliations:** 1Centro de Investigación en Ingeniería y Ciencias Aplicadas, Universidad Autónoma del Estado de Morelos, Campus Chamilpa, Ave. Universidad 1001, Col. Chamilpa, Cuernavaca CP 62209, Mexico; dseuret@uaem.mx (D.S.-J.); eduardo.trutie@uaem.edu.mx (E.T.-C.); ldg@uaem.mx (L.D.-G.); laura.carballo@uaem.edu.mx (L.C.-S.); daily.quintana@uaem.edu.mx (D.Q.-F.); lgaggero@uaem.mx (L.M.G.-S.); 2School of Engineering and Sciences, Tecnologico de Monterrey, Atlixcáyotl 5718, Reserva Territorial Atlix-Cáyotl, Puebla CP 72453, Mexico

**Keywords:** dyadic frequency spectrum, scale–frequency, seismic–acoustic signal, $T_{e}$-gram, $T_{e}$ transform

## Abstract

Using a novel mathematical tool called the $T_{e}$-gram, researchers analyzed the energy distribution of frequency components in the scale–frequency plane. Through this analysis, a frequency band of approximately 12 Hz is identified, which can be isolated without distorting its constituent frequencies. This band, along with others, remained inseparable through conventional time–frequency analysis methods. The $T_{e}$-gram successfully addresses this knowledge gap, providing multi-sensitivity in the frequency domain and effectively attenuating cross-term energy. The Daubechies 45 wavelet function was employed due to its exceptional 150 dB attenuation in the rejection band. The validation process encompassed three stages: pre-, during-, and post-seismic activity. The utilized signal corresponds to the 19 September 2017 earthquake, occurring between the states of Morelos and Puebla, Mexico. The results showcased the impressive ability of the $T_{e}$-gram to surpass expectations in terms of sensitivity and energy distribution within the frequency domain. The $T_{e}$-gram outperformed the procedures documented in the existing literature. On the other hand, the results show a frequency band between 0.7 Hz and 1.75 Hz, which is named the planet Earth noise.

## 1. Introduction

This article presents, through a new mathematical tool called the $T_{e}$-gram, a high-resolution dyadic spectrogram. The authors developed this method and named it $T_{e}$-gram to emphasize its nature as a special type of spectrogram. A spectrogram is a three-dimensional graph that represents the time variation of the energy of the frequency content of a signal. This graph is calculated by analyzing the spectrum within time windows.

This mathematical tool allows the analysis of stochastic signals, as in the case of seismic–acoustic signals [1,2,3]. The $T_{e}$-gram presents a significant advantage compared to the procedures described in the literature. It enables the analysis of energy distribution in a scale–frequency plane using the dyadic frequency spectrum approach. Its benefits include cross-term attenuation, isolation of frequency components, and a multi-sensitive frequency spectrum [4]. Until now, no procedure has been reported that incorporates all these advantages, which makes the $T_{e}$-gram the main novelty presented in this study.

Nowadays, natural-time analysis and time–frequency analysis are widely used for the study of non-stationary stochastic processes. This is because a significant improvement in signal quality is achieved by using natural-time analysis in the time–frequency domain. This technique reduces the uncertainty associated with extracting information from the signal, as shown in previous studies [5,6]. By applying time–frequency analysis, greater precision and clarity in data interpretation are obtained. A prominent example of this is the analysis of seismic–acoustic signals [7,8,9,10]. This type of signal exhibits the aforementioned characteristics and has been extensively investigated using time–frequency analysis [11,12,13]. The study of these acoustic signals is of great interest due to the valuable information they provide about geological structures [7,14,15].

This work proposes a novel mathematical tool by studying an earthquake through its frequency components instead of being limited only to its magnitude. The main objective of this application is to establish a relationship between the natural vibration frequencies of the structures affected by the earthquake. Additionally, it aims to correlate these frequencies with the frequency components present in the seismic–acoustic signal. Through this approach, it is sought to obtain a deeper understanding of the structural response to seismic events.

The frequency components of a seism are of great interest due to their potential to cause significant damage to the structural health of constructions under seismic vibrations. Furthermore, these frequency components also pose a risk to the well-being and safety of living beings [16]. These problems derive from the dynamic relationship between the natural vibration frequency of a structure and the frequency components present in the spectrum of seismic waves [17,18,19]. When these frequencies equalize, the structure goes into resonance, resulting in a considerable increase in deformations and accelerations. Consequently, the structural elements suffer significant stress, which can eventually cause fatigue and deterioration of the structure.

For such reasons, among the researchers, there is a great interest in the precise and effective analysis of the vibrations generated in a seism. The goal is to reveal the characteristics of seismic–acoustic signals through their spectral composition. Among the papers reported in the last few years, the following can be found.

In a study led by researchers in [20], time–frequency analysis was used by employing the spectrogram of seismic signals. The objective of this analysis was to estimate the development of avalanches and frontal velocities. The results were significant, as they effectively predicted avalanches and accurately measured frontal velocities. The authors of [7] conducted research focused on the time–frequency analysis of seismic signals. They employed frequency concentration and time concentration techniques based on the Short-Time Fourier Transform (STFT) in their analysis. These researchers suggest that obtaining a high-quality, time–frequency representation is crucial for revealing local information about seismic signals. This approach enables the description of geological structures based on the analysis of seismic data. In [21], a method is proposed that is aimed at the instantaneous seismic safety of stratified rock slides. In this investigation, the authors use the Hilbert–Huang Transform to extract the characteristics of the seismic signal. The work reported by [22] is aimed at improving the resolution of seismic signals through the secondary time–frequency spectrum based on the S Transform (ST).

The authors in [23,24,25] present a time–frequency analysis of seismic data using a three-parameter ST. These researchers suggest that the Short-Time Fourier Transform (ST) has a low temporal resolution at low frequencies. This limitation hampers the effective extraction of features in the analysis of seismic signals. The main objective of their work was to optimize the time–frequency resolution through modifications in the ST. In [26], a technique based on sparse Bayesian learning is shown to achieve high time–frequency resolution in the analysis of seismic signals. These researchers state that their approach involves using the Ricker wavelet to decompose the seismic signal. Subsequently, they apply the Wigner–Ville distribution as part of their analysis.

The main drawback of the proposal presented by these authors is its dependence on learning, which requires a considerably large volume of data and is difficult to obtain. On the other hand, the study carried out by [27] presents an alternative approach based on the high-order synchronized compression transformation. This alternative method offers remarkable precision in capturing the instantaneous frequency component of seismic signals. These researchers claim to employ a higher-order approximation for amplitude and phase in their analysis. They aim to maximize the concentration of energy in the time–frequency representation of the seismic signals.

In [28], they propose a technique based on the general time-synchronized Chirplet Transform. These researchers suggest that this method allows for obtaining information about frequency-modulated signals. In the research of [29], they present a proposal focused on the detection of seismic signals. Their approach combines time–frequency representation techniques with convolutional neural networks for improved detection accuracy. These authors use the STFT in their proposal. In the investigation presented in [30], the authors conduct a time–frequency analysis of seismic data. They utilize the non-stationary Prony method as the basis for their analysis. These researchers propose calculating the spectrum of the input signal using the Hilbert Transform.

Despite the advances achieved by the scientific community, the work reported so far does not focus on improving the sensitivity of the frequency spectrum. The main drawback of these methods lies in their signal analysis approach based on a time–frequency representation. This perspective limits the ability to obtain a dyadic frequency spectrum to improve spectral sensitivity, attenuate cross-terms, and isolate frequency components. Furthermore, it is not possible to obtain a multi-sensitive frequency spectrum, which makes it impossible to maximize spectral purity.

On the other hand, the work shown in [31] states that different kernel functions have been studied. However, the selection of the best kernel function that solves the problems raised becomes an issue that has not yet been resolved. This paper addresses the previously mentioned issues and aims to bridge the gap in previous works. It achieves this by utilizing a $T_{e}$ kernel as reported in [10], which enables the generation of a scale–frequency representation of the seismic signal under analysis.

The rest of the manuscript is organized as follows. Section 2, entitled “Theoretical background”, presents information on the scientific literature. Section 3, entitled “High-Resolution Dyadic Spectrogram”, presents the main contribution to knowledge for the scale–frequency analysis. Section 4, entitled “Sources of the data used”, shows how in the literature the national seismological system acquired the data. Section 5, entitled “Validation of the $T_{e}$-gram”, corresponds to the results and discussion obtained after analysis of the result. Finally, Section 6 shows the main conclusions of the work.

## 2. Theoretical Background

For the $T_{e}$-gram definition in Section 3, it is necessary to start from the definition of the $T_{e}$ Transform given by the authors in [4,32]. It is also required for the analysis of the seismic signal that will be developed in Section 5.

### 2.1. Heisenberg Time–Frequency Box

A Heisenberg time–frequency box is a dictionary of waveforms used in linear transforms to correlate it with the signal being analyzed. This correlation allows the study of the frequency components present in a stochastic process with a certain resolution. Therefore, the time–frequency resolution is determined with the Heisenberg uncertainty principle [33,34].

If we have a dictionary $D={\left\{\theta_{\gamma }\right\}}_{\gamma \in \Gamma }$ where $\theta \;{}_{\gamma }\in L^{1}\left(R\right)\cap L^{2}\left(R\right)\forall R$ and $\left\|\theta_{\gamma }\right\|=1$, then Equation (1) defines the area of $\theta_{\gamma }$ [4,33].
(1)$$\sigma_{time}\sigma_{frequency}\geq \frac{1}{2}$$
where

γ is a multi-index parameter,

$\sigma_{time}$ defines the time resolution, and

$\sigma_{frequency}$ defines the frequency resolution.

The use of a Heisenberg box in signal analysis, which contains frequency components, offers a key advantage. It enables the study of these components at different resolutions, facilitating the isolation and analysis of individual frequencies. Figure 1 shows the representation of a Heisenberg time–frequency box for $\theta_{t_{n},w_{n}}\forall n\in Z^{+}$.

Figure 1 $t_{n}$ and $w_{n}$ define the time-frequency location. From Figure 1 and Equation (1), it can be observed that, to maintain the area of the Heisenberg box, compensation is required. This means that, if the Heisenberg box contracts in the frequency axis achieving high resolution, it expands in the time axis resulting in low resolution.

### 2.2. Energy Density

The energy density allows us to know how the energy of a signal is distributed in the frequency domain [35,36].

If we have a kernel $\phi \left(t\right)$ centered at t=0, and it is a Heisenberg time–frequency box, then Equation (2) defines the energy density of a signal [35].
(2)$$SF\left[f\left(t\right)\right]\left(\mu ,\eta_{\gamma }\right)={\left|\int_{-\infty }^{\infty }f\left(t\right)\phi_{\mu ,\eta_{\gamma }}^{*}\left(t\right)dt\right|}^{2}$$
where

$SF\left[f\left(t\right)\right]\left(\mu ,\eta_{\gamma }\right)$ is the energy density of $f\left(t\right)$,

μ and $\eta_{\gamma }$ are the time and frequency parameters for the Heisenberg box, respectively, and

γ is a multi-index parameter.

### 2.3. Spectrogram

The spectrogram is a mathematical tool widely used in the analysis of time–frequency signals [37,38,39,40,41]. This tool allows us to calculate the energy density of a signal [35]. The authors of [35] define the spectrogram as shown in Equation (3).
(3)$$Psf\left(\mu ,\omega \right)={\left|\int_{-\infty }^{\infty }f\left(t\right)w_{\mu }\left(t\right)e^{-i\omega t}dt\right|}^{2}$$
where

$Psf\left(\mu ,\omega \right)$ is the spectrogram of $f\left(t\right)$, and

$w_{\mu }\left(t\right)$ is a window-shifted μ.

### 2.4. $T_{e}$ Transform

The $T_{e}$ Transform defined by [4,32] is a mathematical tool that allows for obtaining a dyadic frequency spectrum. Its main advantage is that it allows the isolation of spurious frequency components.

If $f\left(t\right)\in L^{1}\left(R\right)\forall t\in R$, then the $T_{e}$ Transform is defined by [4,32] as shown in Equation (4).
(4)$${\hat{f}}_{DTe}\left[f\left(t\right)\right]\left(\mu ,\xi ,\vartheta \right)=\int_{-\infty }^{\infty }f\left(t\right)g_{\mu ,\xi ,\vartheta }^{*}(t)dt\;\forall \mu ,\xi ,\vartheta \in Z^{+}$$
where

${\hat{f}}_{DTe}\left[f\left(t\right)\right]\left(\mu ,\xi ,\vartheta \right)$ are the coefficients of $f\left(t\right)$,

$g_{\mu ,\xi ,\vartheta }\left(t\right)=w_{\mu }\left(t\right)\psi_{\mu ,\xi }\left(t\right)e^{i2\pi \vartheta t}$ is the $T_{e}$ kernel in $L^{1}(R)\cap L^{2}(R)$,

$\psi_{\mu ,\xi }\left(t\right)=\frac{1}{\sqrt{2^{\xi }}}\psi \left(\frac{t}{2^{\xi }}-\mu \right)$ is a dyadic wavelet function, and

μ and ξ are the scaling and translation parameters, respectively.

## 3. High-Resolution Dyadic Spectrogram

In this section, the authors present the main contribution of this manuscript. It is important to highlight that the mathematical basis of the $T_{e}$-gram was introduced in Section 2.

Currently, signal analysis using the spectrogram energy density calculation is one of the most widely used approaches by the scientific community [42,43,44]. Despite the significant progress achieved in various fields of knowledge using the spectrogram, it has a primary limitation. This limitation pertains to the spectrogram’s inability to perform a dyadic analysis of the frequency components present in a signal.

To solve this problem, the authors of this work, in Section “$T_{e}$-Gram Function”, show a new mathematical tool that does not exist in the state-of-the-art. This tool is the high-resolution dyadic spectrogram, which the authors call $T_{e}$-gram. The main advantage of this tool is that it allows the investigation of the energy density of a signal in the dyadic frequency spectrum.

### $T_{e}$-Gram Function

This subsection shows the main contribution aimed at obtaining the definition of the $T_{e}$-gram through the mathematical basis presented in Section 2.

$T_{e}$-gram provides an energy spectral density for an analysis signal from a stochastic or deterministic system, increasing the sensitivity of the frequency spectrum.

Through reference [4], we know that the function $g_{\mu ,\xi ,\vartheta }(t)$ of Equation (4) is a Heisenberg box. If we replace the function $\theta_{\mu ,\eta_{\gamma }}\left(t\right)$ of Equation (2) with $g_{\mu ,\xi ,\vartheta }(t)$ of Equation (4), we obtain Equation (5) that defines the $T_{e}$-gram.
(5)$$\Lambda \left[f\left(t\right)\right]\left(\mu ,\xi ,\vartheta \right)={\left|\langle f,g_{\mu ,\xi ,\vartheta }\rangle \right|}^{2}={\left|\int_{-\infty }^{\infty }f\left(t\right)g_{\mu ,\xi ,\vartheta }^{*}\left(t\right)dt\right|}^{2}$$
where

$\Lambda \left[f\left(t\right)\right]\left(\mu ,\xi ,\vartheta \right)$ denotes the $T_{e}$-gram, and

${\left|\langle f,g_{\mu ,\xi ,\vartheta }\rangle \right|}^{2}$ is the energy density of $f\left(t\right)$ in the $T_{e}$ domain.

## 4. Sources of the Data Used

The purpose of this section is to provide information about the type of earthquake and the source station of the signal for the validation process. This information will be used in Section 5 for the $T_{e}$-gram.

The signal to be analyzed is of the seism that occurred on 19 September 2017 with a magnitude of 7.1 on the Richter scale. The analysis is performed via the methods reported in the reviewed literature and the $T_{e}$-gram. This signal was acquired under the terms of the Nyquist theorem using a sample frequency of 100 Hz. This seism occurred inside the Cocos oceanic plate (i.e., interplate seism) between the states of Morelos and Puebla belonging to the United Mexican States. The 19 September seism caused significant human and material losses to urban communities, including a large area of Mexico City [45].

Seismic events that occur in the United States of Mexico with epicenters off the Pacific coast are typically subduction earthquakes. These earthquakes are generated due to the interaction between the Cocos and North American tectonic plates with the interplate type being less common in this region. On 19 September 2017, an interplate earthquake occurred resulting in extensive damage to structures in the state of Morelos, Mexico. The epicenter of this earthquake was located at coordinates 18.40 N latitude and −98.72 W longitude with a depth of 57 km. For this reason, the authors of this work decided to use the signal of this seism for the validation process of the proposed methodology.

The seismic signals used in the analysis are sourced from the Yautepec station (YAIG), which is part of the wideband network of the National Seismological Service of the United Mexican States. The seismic signals used come from the Yautepec station (YAIG, belonging to the wide band network of the National Seismological Service of the United Mexican States) because it is the closest to the epicenter of the earthquake. In this study, the HLZ (vertical) component of the three components reported by the seismological station was used [46].

## 5. Validation of the *$T_{e}$*-Gram

To carry out the validation process, three different scenarios were designed. In the first scenario, the signal is analyzed before the earthquake occurs. In the second scenario, the signal is examined throughout the entire duration of the earthquake. Finally, in the third scenario, the signal after the earthquake is analyzed.

Figure 2 shows the signal of the earthquake to be analyzed in the time domain. On the other hand, Figure 2a depicts the analyzed signal, accentuating the specific segments earmarked for the study. Furthermore, Figure 2b shows the zooming of the signal during the seism.

In the signal processing with the $T_{e}$-gram, the authors utilized a Daubechies 45 wavelet function ($\psi_{\mu ,\xi }\left(t\right)$). The reason for choosing this particular wavelet function is its ability to provide an attenuation close to 150 dB for the rejection band, as reported in [47]. Additionally, a Kaiser-6 window ($w_{\mu }\left(t\right)$) was employed [4].

Figure 3, Figure 4, Figure 5, Figure 6 and Figure 7 show the results obtained after applying the methods reported in the reviewed literature to the signal shown in Figure 2a. Note how the methods reported in the literature are based mainly on a time–frequency analysis.

Figure 3a–c presents the findings after processing the signal from Figure 2a. To carry out this processing, the authors used a mathematical tool called spectrogram.

The tool used for processing the seismic signal, as depicted in Figure 2a, was the Hilbert–Huang Transform. The results of this transformation are presented in Figure 4a (before the seism), Figure 4b (during the seism), and Figure 4c (after the seism). The results with the Hilbert–Huang Transform do not show the presence of frequency components before and after the earthquake. This limitation hinders the analysis of small changes that may occur before and after the seismic event.

The third validation process was carried out via the S Transform. The results presented in Figure 5a–c reveal the behavior of the seism before, during, and after, respectively.

Figure 6a (before the seism), Figure 6b (during the seism), and Figure 6c (after the seism) depict the results after processing the signal shown in Figure 2a. This processing was performed via the Wigner–Ville distribution. Observe how in Figure 6a a slight variation is perceived before the seism.

Figure 7a (before the seism), Figure 7b (during the seism), and Figure 7c (after the seism) show the results after processing the signal from Figure 2a. The mathematical tool used in this processing was the Chirplet Transform. Observe how in Figure 7a a slight variation is perceived before the seism.

The results obtained with the $T_{e}$-gram proposed in this work is presented in Figure 8 (see Figure 8a–c). In Figure 8a, it can be observed that the energy of the signal, both before and after the earthquake, is concentrated at scale 6. This concentration is due to its association with a frequency band ranging from 0.75 Hz to 1.35 Hz. It is important to note that this result is not observed in the methods reported in the literature. However, during the occurrence of the earthquake, as depicted in Figure 8b1, it can be observed that the energy is concentrated at scale 2. This distinct behavior indicates a frequency band that varies from 12 Hz to 25 Hz. After analyzing the results, it is evident that the $T_{e}$-gram surpasses the methods reported in the state-of-the-art.

From Figure 8a–c, a frequency band is observed that begins at 12 Hz throughout the seism. This frequency band is very hard to isolate and observe in the analysis performed with the mathematical tools reported in the literature (see results in Figure 3b, Figure 4b, Figure 5b, Figure 6b and Figure 7b).

Figure 8a–c reveals a phenomenon of great interest related to the seismic process. In scale 6, a notable presence of low-frequency vibrations and a high concentration of energy are observed, which could be the origin of the said process. In the Mexican Pacific, two conducive conditions converge to generate this phenomenon. First, faults occur as a result of the discontinuity between the continental plates. Second, the presence of coasts serves as a boundary between the sea and the continent. In this area, sea waves impact the continent with low frequency, transferring energy to the mountain ranges located between the plates.

The energy accumulated in these low-frequency vibrations can be equated to the charge of a capacitor. When a resistance threshold is reached between the irregularities of the plates, an uncontrolled sliding between them is triggered, generating an earthquake. In this capacitor model, the accumulated energy exceeds the potential energy accumulated in the irregularities thus triggering the discharge, that is, the earthquake. The band that we refer to is 0.75–1.75 Hz, and we called it “planet Earth noise”.

## 6. Conclusions

This paper presents a new methodology, introducing a new mathematical tool called $T_{e}$-gram.

This tool allows us to know the spectral density of a signal’s energy in a scale–frequency plane. This mathematical tool is especially suitable for the study of seismic signals, which are multi-component, non-stationary, and variable in time.The $T_{e}$-gram offers an effective solution to address these features and provides a unique perspective in seismic signal analysis.The $T_{e}$-gram, which is based on the $T_{e}$ Transform, enhances spectral sensitivity.Additionally, the $T_{e}$-gram successfully reduces cross-terms, isolates frequency components, and yields a multi-sensitive frequency spectrum. This methodology maximizes spectral purity and has proven to be highly effective for this purpose.An important advantage is a demonstration that the variation in the $T_{e}$ kernel provides a higher sensitivity in the way of knowing the information of the frequency spectrum. This essential feature enables a better understanding and detailed analysis of signals. This crucial feature facilitates a better understanding and more detailed analysis of signals, thereby significantly enhancing the ability to extract precise information from the frequency spectrum.Through this method, it is possible to identify a frequency band between 12 Hz and 24 Hz. This band was not perceptible in the other mathematical tools, as shown in Figure 3a, Figure 4, Figure 5, Figure 6 and Figure 7c). This identification has facilitated the establishment of a relationship between the natural frequency vibration of civil structures, typically at approximately 12 Hz, and the frequency components of the second scale of seismic signals during an earthquake. The authors of this study consider this finding as the main cause of the serious damage that occurs in buildings during an earthquake.

This work proved that the $T_{e}$-gram exhibits greater effectiveness compared to the methods previously reported in the literature and reviewed by the authors. In future studies, a theoretical investigation will be conducted to explore the energy conservation aspects utilizing the $T_{e}$-gram and the study of Earth noise.

## Figures and Tables

**Figure 1 sensors-23-06051-f001:**
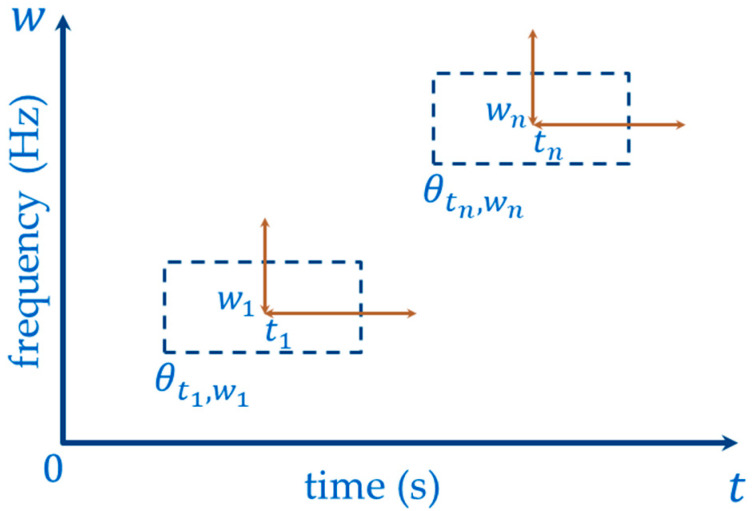
Heisenberg box representation for $\theta_{t_{n},w_{n}}$ Source: Own elaboration.

**Figure 2 sensors-23-06051-f002:**
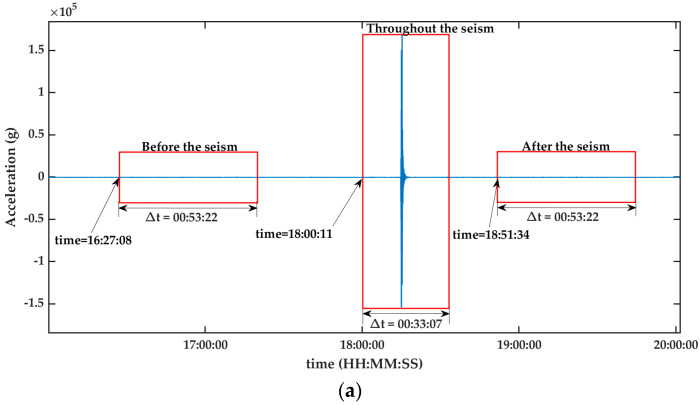
Displays the seismic signal in the time domain. (**a**) The analyzed signal encompasses the different temporal moments related to the seism. This covers both the period before the seismic event, during the seism itself, and also the period after it. Δt represents the time of each box. (**b**) Zooming of the signal during the seism is shown in (**a**). Source: Own elaboration.

**Figure 3 sensors-23-06051-f003:**
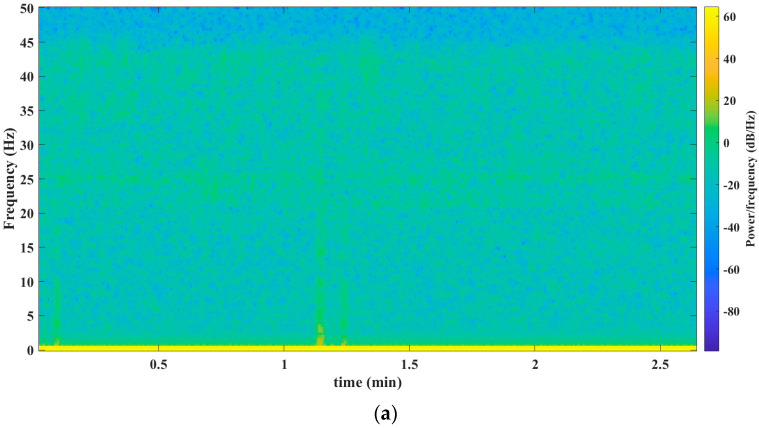
Results obtained from the spectrogram mathematical tool. (**a**) Spectrogram before the seism. Spectrogram throughout the seism. The precise timestamp of the data’s start for analysis is 16:27:08 (HH:MM:SS). (**b**) The precise timestamp of the data’s start for analysis is 18:00:11 (HH:MM:SS). (**c**) Spectrogram after the seism. The precise timestamp of the data’s start for analysis is 18:51:34 (HH:MM:SS). Source: Own elaboration.

**Figure 4 sensors-23-06051-f004:**
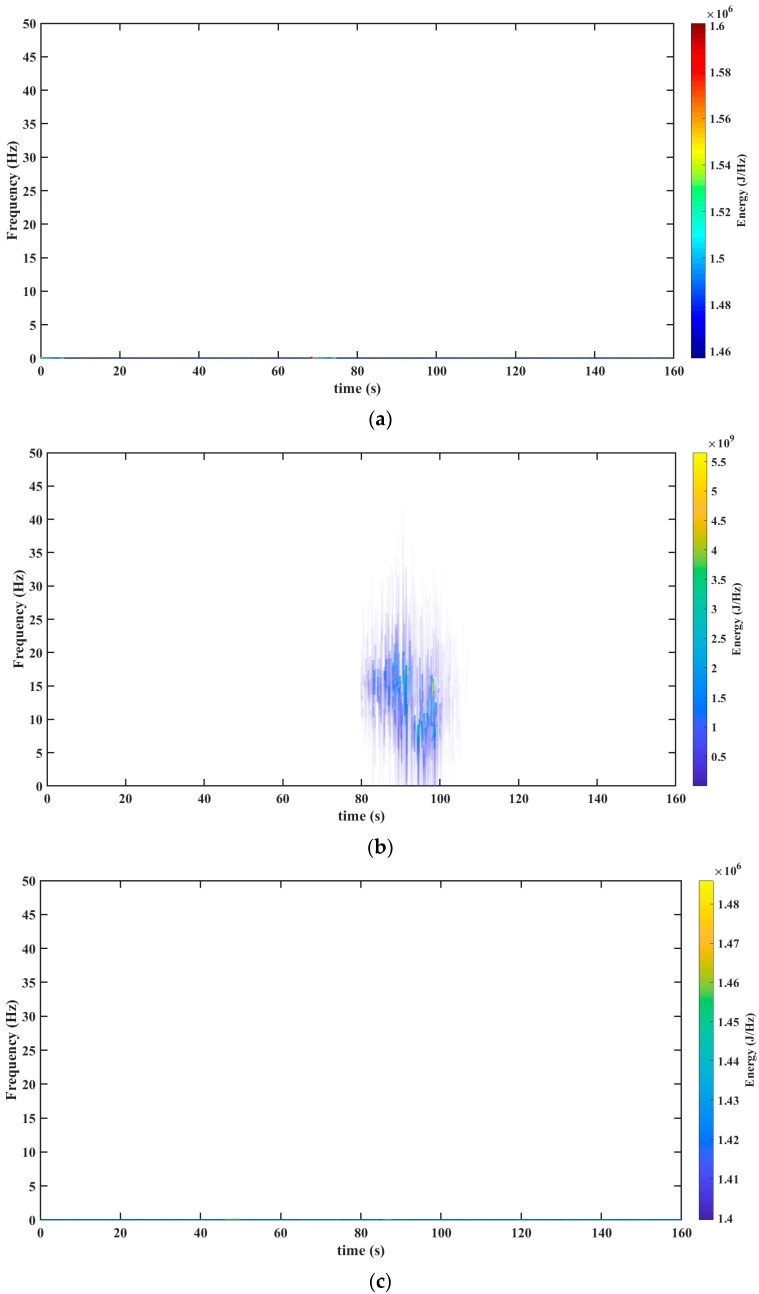
Results obtained from the Hilbert–Huang Transform mathematical tool. (**a**) Hilbert–Huang Transform before the seism The precise timestamp of the data’s start for analysis is 16:27:08 (HH: MM: SS). (**b**) Hilbert–Huang Transform throughout the seism. The precise timestamp of the data’s start for analysis is 18:00:11 (HH: MM: SS). (**c**) Hilbert–Huang Transform after the seism. The precise timestamp of the data’s start for analysis is 18:51:34 (HH: MM: SS). Source: Own elaboration.

**Figure 5 sensors-23-06051-f005:**
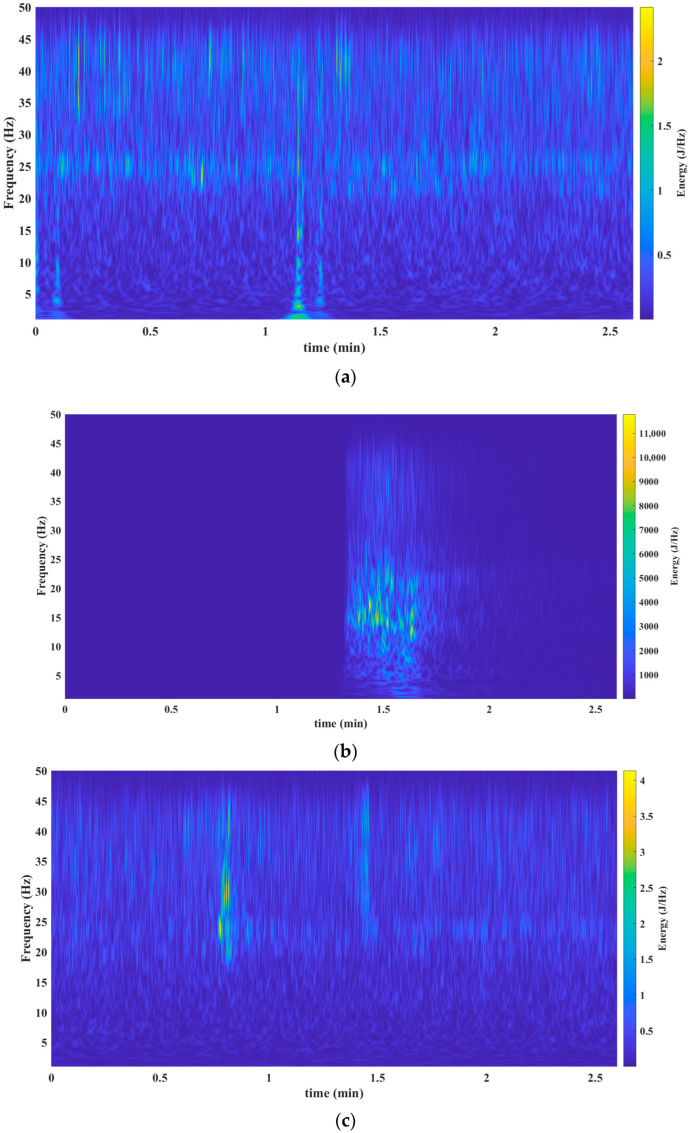
Results obtained from the mathematical tool S Transform. (**a**) S Transform before the seism. The precise timestamp of the data’s start for analysis is 16:27:08 (HH: MM: SS). (**b**) S Transform throughout the seism. The precise timestamp of the data’s start for analysis is 18:00:11 (HH: MM: SS). (**c**) S Transform after the seism. The precise timestamp of the data’s start for analysis is 18:51:34 (HH: MM: SS). Source: Own elaboration.

**Figure 6 sensors-23-06051-f006:**
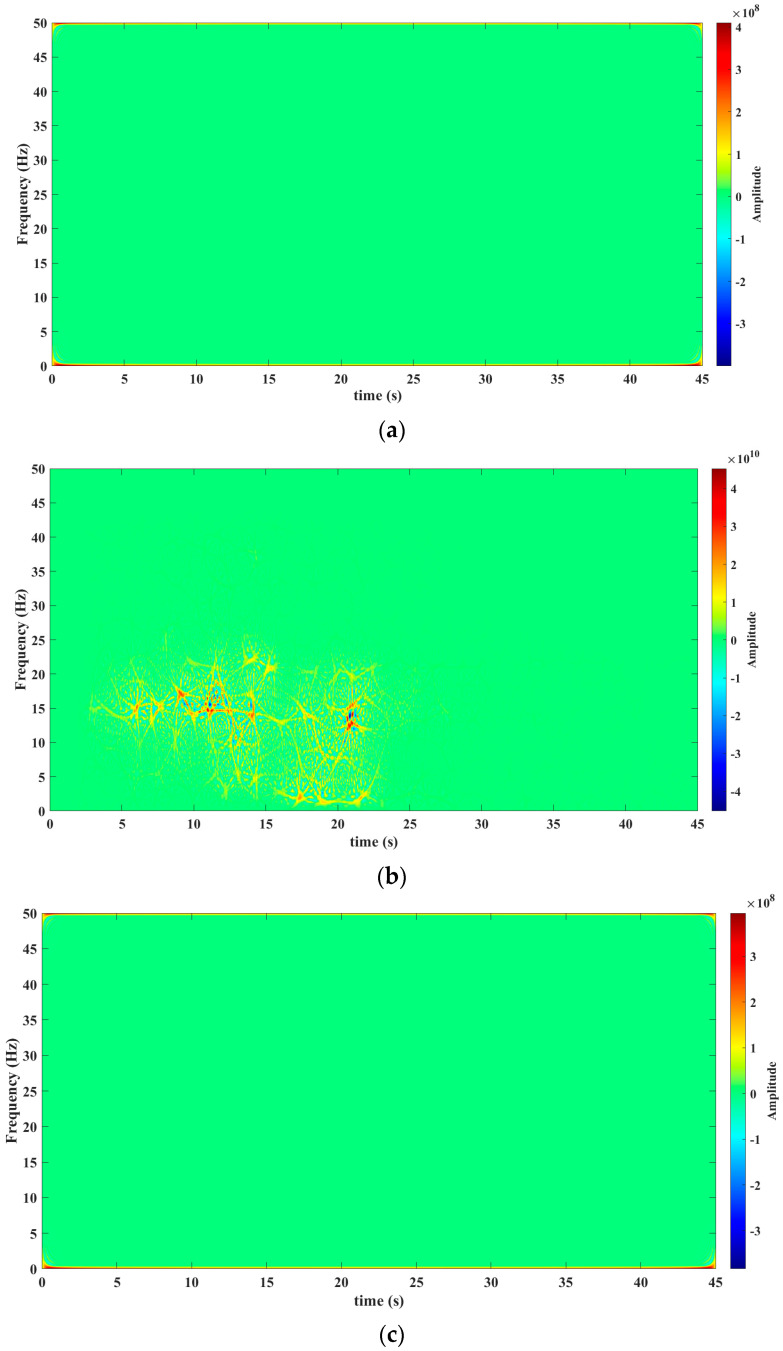
Results obtained from the Wigner–Ville distribution mathematical tool. (**a**) Wigner–Ville distribution before the seism. The precise timestamp of the data’s start for analysis is 16:27:08 (HH: MM: SS). (**b**) Wigner–Ville distribution throughout the seism. The precise timestamp of the data’s start for analysis is 18:00:11 (HH: MM: SS). (**c**) Wigner–Ville distribution after the seism. The precise timestamp of the data’s start for analysis is 18:51:34 (HH: MM: SS). Source: Own elaboration.

**Figure 7 sensors-23-06051-f007:**
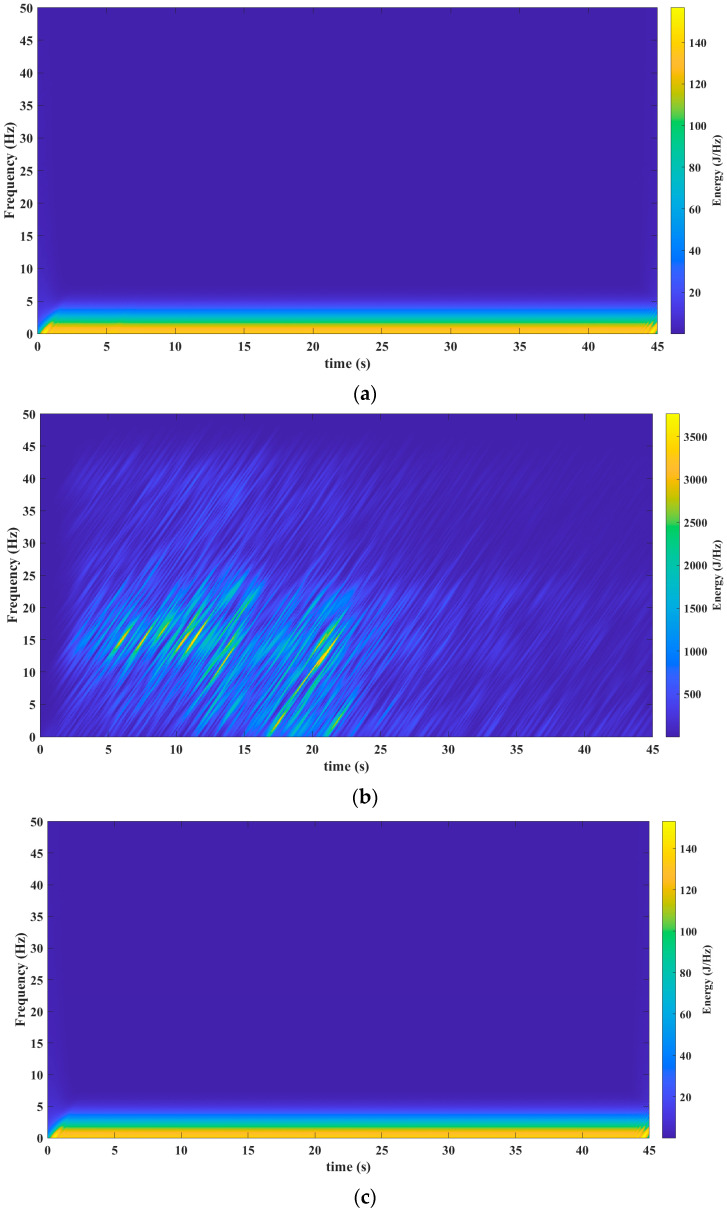
Results obtained from the mathematical tool Chirplet Transform. (**a**) Chirplet Transform before the seism. The precise timestamp of the data’s start for analysis is 16:27:08 (HH: MM: SS). (**b**) Chirplet Transform throughout the seism. The precise timestamp of the data’s start for analysis is 18:00:11 (HH: MM: SS). (**c**) Chirplet Transform after the seism. The precise timestamp of the data’s start for analysis is 18:51:34 (HH: MM: SS). Source: Own elaboration.

**Figure 8 sensors-23-06051-f008:**
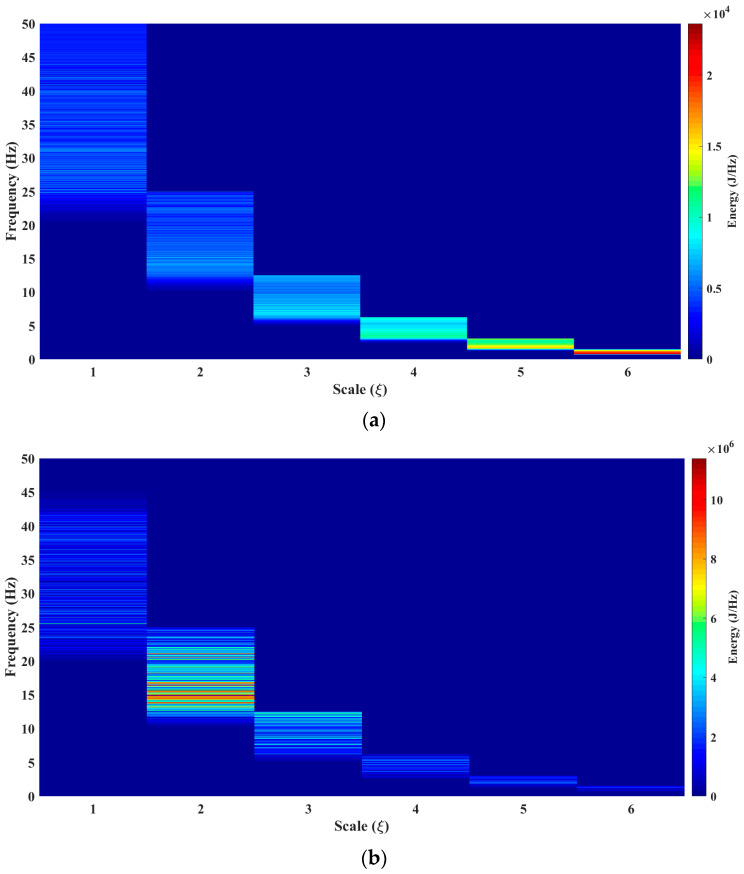
Results obtained from the mathematical tool $T_{e}$-gram. (**a**) $T_{e}$-gram before the seism. (**b**) $T_{e}$-gram throughout the seism. (**b1**) $T_{e}$-gram when ξ=2 throughout the seism. (**c**) $T_{e}$-gram after the seism.

## Data Availability

SSN (año): Universidad Nacional Autónoma de México, Instituto de Geofísica, Servicio Sismológico Nacional, México. website: http://www.ssn.unam.mx (accessed on 4 February 2023).

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
