# Peer review of "Feature Extraction of a Non-Stationary Seismic–Acoustic Signal Using a High-Resolution Dyadic Spectrogram"

_sensors, 2023, doi:10.3390/s23136051_

Round 1

Reviewer 1 Report

The presentation adopted by the authors are not suitable. The section 5 related to the validation is the main section of the study and the authors have not explained the validated figures in detailed form. All the figures are attached in a continuous manner without explanations. There are mere 15 lines in this section. Above that, in section 2 and 3, basics are written which are related to the literature. Section 4 is related to the data employed in the study. Overall the study is not presented in a professional manner and does have a deep influence on the international audience. Therefore, the study in the current form is not acceptable.

Reviewer 2 Report

In this manuscript (ms), the authors present a method of analysis of real time, non-stationary signals using a high-resolution dyadic spectrogram. They employ a novel mathematical tool called T_e-gram (introduced by the first three authors in Ref.[4]) for analyzing the energy distribution of frequency components in the scale-frequency plane. As an example of a non-stationary seismic-acoustic signal, the authors study the vertical component of the seismogram recorded at the Yautepec station before, during and after the M7.1 earthquake (EQ) that occurred on 19 September 2017 the epicenter of which was located in the Mexican flat slab, close to Mexico City, causing important human losses and significant material damages around downtown area [Flores-Márquez et al., Entropy, Vol. 22 (2020), 730]. The ms is well structured, but the presentation needs significant improvement before publication. Since the results presented are very interesting and potentially provide explanations for the extensive damage caused by this M7.1 EQ, I consider that this research deserves publication in SENSORS and the authors should submit a revised version of the ms in which they will have improved the following important presentation points:

1)SENSORS is a journal of interdisciplinary audience. Hence, the term “Heisenberg time-frequency box” should be defined explicitly.

2)The absence of typing mistakes in the Equations is of valuable importance for the authors to convey their original ideas to the readers. Thus, an exponent “2” should be inserted into rightmost part of Eq.(1) and in line 133 the text should read “a window shifted by \mu.” In the same context, \psi_{\mu,\xi}(t) should be defined in Section 2.

3)In the second paragraph of page 5, the authors discuss the properties of the 19 September 2017 M7.1 EQ on the Mexican flat slab. For the readers’ better information, the paper

Flores-Márquez, E.L.; Ramírez-Rojas, A.; Perez-Oregon, J.; Sarlis, N.V.; Skordas, E.S.; Varotsos, P.A. Natural Time Analysis of Seismicity within the Mexican Flat Slab before the M7.1 Earthquake on 19 September 2017. Entropy 2020, 22, 730. https://doi.org/10.3390/e22070730

should be also mentioned there since they provide essential details about this EQ.

4)The boxes drawn in Fig. 2 should correspond to the time intervals analyzed by the various techniques in Figs. 3 to 21, otherwise the reader cannot see the connection of the real time signal of Fig.2 to the spectra depicted. Moreover, the reader would like to see in Fig.2 a zoom in the case of the “during the EQ” window so that he/she can evaluate the success of the various methods employed.

5)In Figs. 3 to 17, in each figure caption the exact time of the beginning of the data analyzed in term of the time scale reported in Fig.2 should be given. Additionally, reference to the original literature that includes the exact formulas used for each transform should be also mentioned.

6)If I do not miss something on the time scale (see point 6, above), the Wigner-Ville function (Fig. 13) reveals a pre-seismic variation. Is this an artifact or a true phenomenon? The same applies to the Chirplet Transform (Fig. 16). The authors should anyhow discuss this observation.

7)In Figs. 18 to 21, the authors present their findings in terms of the scale \xi, however, no physical unit for this scale is defined throughout the ms. This is very important for the evaluation of the suggested method. Please provide the appropriate units.

8)In line 302, Data Availability Statement: I am afraid of a misunderstanding since I don’t know what “ethical of privacy restrictions” apply to the vertical component of a (probably publicly available) seismogram. Moreover, the need for reproducibility of the results presented, so that the proposed method can be also applied by other scientists, requires more details on the selection of functions w_\mu(t) and \psi_{\mu,\xi}(t)  involved in the T_e transform of Eq.(3). The authors should revise their “Data Availability Statement”.

9) The ms would benefit from the elimination the typing errors: a) In line 111 “Section 4 explains how”, b)In line 268 “we called it "planet Earth noise"”, and c)In line 314, there is no author or bibliographic information, just the title exists in Ref.[3].

In view of the above, I consider that the authors should revise their ms and resubmit.

Reviewer 3 Report

The authors of this manuscript (ms) present a novel method for feature extraction of non-stationary signals which is based on a high-resolution dyadic spectrogram. This topic is very interesting and such a contribution may be proved very important. Although the paper is well-written, the presentation needs to be improved before publication.

Especially, in the third paragraph of the Introduction, the authors discuss the importance of time-frequency analysis in modern applications which include seismology. They do not mention, however, natural time analysis on the foundation of which is time-frequency analysis, see, e.g., Chapter 2.6 of [R1], which has been extensively applied to the study of earthquakes and related models, see [R2] and references therein. For the readers’ better understanding the author should mention natural time analysis in this paragraph (i.e., at lines 40-45) by referring to the following citations:

R1. Varotsos P.A., Sarlis N.V. and Skordas E.S., Natural Time Analysis: The new view of time. Precursory Seismic Electric Signals, Earthquakes and other Complex Time-Series (Springer-Verlag, Berlin Heidelberg) 2011. https://doi.org/10.1007/978-3-642-16449-1

R2. N.V. Sarlis, E.S. Skordas, and P.A. Varotsos, Similarity of fluctuations in systems exhibiting Self-Organized Criticality, EPL 96 (2011) 28006. https://doi.org/10.1209/0295-5075/96/28006

In summary, I will be glad to suggest publication of a ms revised along the lines mentioned above.

Minor editing of English language required

Round 2

Reviewer 1 Report

Authors have provided adequate responses to my queries. The study is an improved version. Some minor suggestions are recommended prior to publication.

1. Provide the sections that shows which section belongs to literature, data and methods, and results & discussion.

2. write standout conclusions in bullet points.

3. check the English language carefully.

4. Ignore writing 2-3 lines paragraph.

5. Some figures in section 5 can be merged into sub-figures to make it a concise study.

3. check the English language carefully.
